# The effectiveness of coagulation sink of 3-10 nm atmospheric particles

Runlong Cai[1,*], Ella Häkkinen[1], Chao Yan[1,2], Jingkun Jiang[3], Markku Kulmala[1,2], Juha Kangasluoma[1,4]

[1]Institute for Atmospheric and Earth System Research / Physics, Faculty of Science, University of Helsinki, Helsinki, 00014, Finland

[2]Aerosol and Haze Laboratory, Beijing Advanced Innovation Center for Soft Matter Science and Engineering, Beijing University of Chemical Technology, Beijing, 100029, China

[3]State Key Joint Laboratory of Environment Simulation and Pollution Control, School of Environment, Tsinghua University, Beijing, 100084, China

[4]Karsa Ltd., A. I. Virtasen aukio 1, Helsinki, 00560, Finland

*Correspondence to:* Runlong Cai (runlong.cai@helsinki.fi)

**Abstract.** As a major source of ultrafine particles, new particle formation (NPF) occurs frequently in various environments. However, the survival of new particles and the frequent occurrence of NPF events in polluted environments have long been perplexing, since new particles are expected to be scavenged by high coagulation sinks. Towards solving these problems, we establish an experimental method and directly measure the effectiveness of the size-dependent coagulation sink of monodisperse 3-10 nm particles in well-controlled chamber experiments. Based on the chamber experiments and long-term atmospheric measurements from Beijing, we then discuss the survival of new particles in polluted environments. In the chamber experiments, the measured coagulation coefficient increases significantly with a decreasing particle size, whereas it is not sensitive to the compositions of test particles. Comparison between the measured coefficient with theoretical predictions shows that almost every coagulation leads to the scavenging of one particle, and the coagulation sink exceeds the hard-sphere kinetic limit due to van der Waals attractive force. For urban Beijing, the effectiveness of coagulation sink and a moderate or high (e.g., > 3 nm h[-1]) growth rate of new particles can explain the occurrence of measured NPF events; the moderate growth rate further implies that in addition to gaseous sulfuric acid, other gaseous precursors also contribute to the growth of new particles.

## 1 Introduction

New particle formation (NPF) is a major source of atmospheric aerosol number concentration, during which gaseous precursors form new particles via nucleation (Kulmala et al., 2012a). A fraction of these new particles can grow up to the cloud condensation nucleus size (e.g., > 50 nm) and influence the climate (Kuang et al., 2009; Gordon et al., 2017). Some studies also report that new particles may contribute to haze formation (Guo et al., 2014; Kulmala et al., 2021). The influences of NPF are governed by the number of new particles formed via nucleation and their survival probabilities. In the past decade,



instrument development (Jiang et al., 2011; Kulmala et al., 2013), atmospheric measurements (Chen et al., 2012; Bianchi et

al., 2016; Sipilä et al., 2016; Yao et al., 2018; Lehtipalo et al., 2018), laboratory experiments (Kirkby et al., 2011; Kürten et

al., 2014; Riccobono et al., 2014; He et al., 2021; Jen et al., 2014), and theoretical studies (Ortega et al., 2012; Yu et al., 2018)

have led to significant advances in the understanding of nucleation. However, particle survival during the growth process is

still poorly understood and under debate. Since the concentration of grown particles is more sensitive to survival probability

than nucleation rate (Westervelt et al., 2014), understanding the growth process and accurately determining the survival

probability are pivotal to assessing the influences of NPF on regional air quality and global climate.

One of the most confusing problems related to particle survival is that even after accounting for the high nucleation rate in

polluted environments (Cai and Jiang, 2017; Yao et al., 2018; Cai et al., 2021c), how could new particles survive and be

frequently observed (Kulmala et al., 2017) at a high aerosol surface area concentration? One straightforward approach to

address this problem is to retrieve particle survival probability from the evolution of particle size distributions (Weber et al.,

1997; Kuang et al., 2009; Kulmala et al., 2017) and compare it to theoretical predictions. Such a comparison reduces the

complexity of the problem, yet it does not provide the reasons why the retrieved and theoretical survival probabilities

sometimes deviate from each other (Kulmala et al., 2017). Further, both the retrieved and theoretical survival probabilities

may be very sensitive to measurement uncertainties and non-homogeneity of the atmosphere, which presents a challenge to

interpreting the comparison results.

Alternatively, one can examine the accuracy of the parameters determining the survival probability via direct measurements.

According to theoretical derivations (Weber et al., 1997; Kerminen and Kulmala, 2002; McMurry et al., 2005), particle survival

probability is determined by the ratio of their net growth rate (GR) to their loss rate characterized by the coagulation sink

(CoagS), i.e., CoagS/GR. Hence, a bias in the theoretically predicted survival probability must be caused by biases in CoagS

and/or GR. While the uncertainty range of GR retrieved from the measured particle size distributions could be estimated, the

effectiveness of the theoretically calculated CoagS was questioned. For example, to explain the observed frequent NPF events

in polluted megacities, Kulmala et al. (2017) hypothesized that the CoagS was overestimated due to unconsidered ineffective

coagulation between new particles and large particles (i.e., scavengers).

Hence, although challenging, accurate determination of the coagulation coefficient between new particles and large particles

using well-controlled experiments is fundamental to estimating particle survival probabilities and the influences of NPF. There

have been studies measuring the coagulation coefficient. Okuyama et al. (1984) and Okuyama et al. (1986) measured the

coagulation coefficients of NaCl, ZnCl$_2$, and Ag nanoparticles using a long metal pipe. They observed effective coagulation

among these particles and reported significant contributions from van der Waals force to the coagulation coefficient. Similar

findings for $H_2SO_4$-$H_2O$ particles were reported in Chan and Mozurkewich (2001), in which study the coagulation coefficient of monodisperse particles was determined using a 5.2 L tank reactor. Ineffective coagulation was often reported for combustion-generated nascent soot particles (D'Alessio et al., 2005; Sirignano and D'Anna, 2013; Hou et al., 2020). These previous experiments were mainly focused on the coagulation between nanoparticles of similar size. The coagulation coefficients were estimated according to the change in the measured particle size distribution. For atmospheric NPF, however, CoagS is governed by the coagulation coefficients between small new particles and large (e.g., 100 nm) scavengers (Cai et al.,

2017b). As a result, the effectiveness of the CoagS of new particles remains unknown, and its determination requires new methods and apparatus to minimize the change of particle size distribution due to self-coagulation between new particles.

In this study, we provide direct experimental evidence for the effectiveness of the CoagS of new particles. An experimental method is established to determine the size-dependent coagulation coefficient of monodisperse nanoparticles from their steady-state concentration and decay rate in chamber experiments. Using this method, we measure the coagulation coefficient of 3-

10 nm particles composed of $NH_4HSO_4$, NaCl, and oxygenated organic molecules (OOMs). These test particles are used to represent atmospheric new particles formed by acid-base nucleation and organics nucleation. The sub-10 nm size range is selected because the scavenging of new particles is most significant in this range. The effectiveness of CoagS is then demonstrated by comparing the measured coagulation coefficients to theoretical values. Based on the measured effectiveness of CoagS, we further explore the potential reasons for frequent intensive NPF events using atmospheric data measured in urban

Beijing.

## 2 Methods

### 2.1 Laboratory Experiment

The size-dependent CoagS of 3-10 nm particles was experimentally determined using an environmental chamber. This

chamber was made from Teflon and its volume was 1 $m^3$. The total flow rate through the chamber was 10 L $min^{-1}$ and hence the residence time was ~100 min. A low-speed fan was used to mix particles in the chamber. Experiments were conducted at room temperature (~20 °C) and we note that the coagulation coefficient is not sensitive to temperature. Before each experiment, the chamber was flushed using filtered compressed air to remove the remaining particles.

As shown in Figure 1, 100 nm $NH_4HSO_4$ and NaCl particles were generated and injected into the chamber to provide the

CoagS of sub-10 nm particles. The size of 100 nm was selected because particles around this size contribute most to CoagS in urban environments (Cai et al., 2017b). Monodisperse sub-10 nm $NH_4HSO_4$, NaCl, and OOM particles were also injected into





the chamber. Among the test particles, $NH_4HSO_4$ particles represent atmospheric new particles formed by acid-base nucleation (Chen et al., 2012; Yao et al., 2018); OOM particles represent new particles formed by nucleation with highly oxygenated organic molecules and particles coated by OOMs during their growth (Bianchi et al., 2016; Tröstl et al., 2016); and NaCl

particles were used to keep consistency with previous studies (Okuyama et al., 1984, 1986) on the coagulation between nanoparticles with similar size. We generated these test particles outside the chamber instead of producing them within the chamber. In this way, we avoided the perturbation of significant vapor condensation on particle evolution during the experiments.

===================

95              Figure 1 (double column)

===================

$NH_4HSO_4$ and NaCl particles were generated in a tube furnace using the condensation-evaporation technique (Scheibel and Porstendörfer, 1983; Kangasluoma et al., 2013) (Figure 1). They were charged and classified using a Vienna-type differential mobility analyzer (DMA). The aerosol and sheath flow rates of the DMA were 3.5 and 10 L min$^{-1}$, respectively, to maximize

the classified particle concentrations. A neutralizer was used to neutralize the classified charged particles before they entered the chamber. An aerosol electrometer (Fernandez de la Mora et al., 2017) (SEADM S.L.) was used to monitor the classified particle concentration. Two mass flow controllers were used to control the classified particle concentration by adjusting the dilution ratio so that the particle concentration entering the chamber could be kept stable.

OOM particles were generated in a flow tube from ozone-initiated limonene oxidation products (Figure 1). The inlet flow of

the flow tube was made up of 2 L min$^{-1}$ dry zero air and 1 L min$^{-1}$ humidified zero air. A tiny amount of limonene was added to the inlet flow via diffusion from a syringe tip. By doing this, we minimized the amount of OOMs and hence the size of OOM particles. Ozone was generated in the flow tube using an ultraviolet-C lamp. The influence of OH radicals on OOMs generation was supposed to be minor, though it was not critical for this experiment. OOMs nucleated into particles in the flow tube and the remaining gas-phase OOMs downstream of the flow tube were removed by a denuder containing activated carbon.

In addition to the denuder, an open-loop DMA also contributed to separating particles from gas-phase OOMs such that no new particles were formed in the chamber. The generated OOM particles were volatile and they evaporated slowly in the vapor-free environment in the chamber. However, this slow evaporation did not affect particle number concentration and the change of particle size was minor (< 20 %) during each experimental run.

Large $NH_4HSO_4$ and NaCl particles were generated using an atomizer. The generated particles were dried and then classified

using a DMA with a centroid diameter of 100 nm. Classified particles were neutralized and subsequently injected into the





chamber. The concentration of 100 nm particles entering the chamber was monitored by a condensation particle counter (CPC; model 3750, TSI Inc.) and dynamically adjusted by two mass flow controllers, so that the self-coagulation of 100 nm particles was negligible whereas the CoagS of sub-10 nm particles contributed by these 100 nm particles could be well measured against wall loss and dilution.

Particle concentrations in the chamber were measured using a scanning mobility particle spectrometer (SMPS; model 3936, TSI Inc.) and a CPC (model 3772, TSI Inc.). The SMPS was mainly used to measure the size distributions of sub-10 nm particles though it could also cover the distribution of 100 nm particles. When measuring the CoagS of small (e.g., 3-5 nm) particles, the scan range was focused on the sub-10 nm particle size to obtain good statistics. The CPC was used to measure total particle concentration in the chamber, which was approximately equal to 100-nm particle concertation because of the

high wall loss of sub-10 nm particles.

The CoagS of sub-10 nm particles were determined using a steady-state concentration approach and a decay rate approach. For the steady-state concentration approach, we measured the pseudo-steady-state concentration of sub-10 nm particles as a function of 100 nm particle concentration. This approach benefited from a large number of data points for each experimental run, which overcame the uncertainties associated with measurements and fitting. For the decay rate approach, the CoagS was

derived using the enhancement in the decay rates of sub-10 nm particles due to the CoagS contributed by a certain concentration of 100 nm particles. This approach has previously been used to measure the mass accommodation coefficient of volatile organic compounds (Krechmer et al., 2017). Here we used it for OOM particles whose concentration could not be kept stable for hours as required by the steady-state concentration approach.

**2.2 Atmospheric measurements**

Atmospheric NPF events used in this study were measured in Beijing from Jan 16, 2018, to Dec. 31, 2019. The measurement site was located at the west campus of Beijing University of Chemical Technology (Scheibel and Porstendörfer, 1983). 1.5 nm – 10 μm particle size distributions were measured using a homemade diethylene glycol SMPS (1.5 – 6 nm) (Cai et al., 2017a) and a particle size distribution system (3 nm – 10 μm) (Liu et al., 2016). NPF days with intensive sub-3 nm particle formation and subsequent new particle growth were classified based on the measured size distributions (Deng et al., 2020). We use the

occurrence of NPF events to characterize particle survival to avoid the potentially large uncertainties in the survival probability retrieved from particle size distributions. The gaseous sulfuric acid concentration was measured using a nitrate chemical ionization time-of-flight mass spectrometer (Aerodyne Inc.) (Lu et al., 2019). The condensation sink (CS) of sulfuric acid during NPF periods was calculated using the measured particle size distributions. Detailed information on the measurement site and the dataset can be found elsewhere (Deng et al., 2020; Deng et al., 2021).





**2.3 Box model**

A box model based on aerosol kinetics is used to simulate the evolution of particle concentration in the chamber. The governing population balance equation for sub-10 nm particle concentration is

$$\frac{dN_{sub10}}{dt} = Q - [\beta(d_p)N_{100} + WL(d_p) + DL] \cdot N_{sub10} \tag{1}$$

where $N_{sub10}$ and $N_{100}$ are the concentrations (cm$^{-3}$) of monodisperse sub-10 nm and 100 nm particles, respectively; $t$ is time (s); $Q$ is the source rate (cm$^{-3}$ s$^{-1}$) of sub-10 nm particles resulting from particle injection into the chamber; $\beta(d_p)$ is the

coagulation coefficient (cm$^3$ s$^{-1}$) between sub-10 nm and 100 nm particles; $d_p$ is the diameter (nm) of sub-10 nm particles; $WL(d_p)$ is the size-dependent wall loss rate (s$^{-1}$) of sub-10 nm particles; and DL is dilution rate (s$^{-1}$). The size-dependent CoagS of sub-10 nm particles is equal to $\beta(d_p)N_{100}$.

Setting the $dN_{sub10}/dt$ in Eq. 1 to zero yields the steady-state value of $N_{sub10}$ ($N_{sub10}^{ss}$) as a function of $d_p$ and $N_{100}$,

$$N_{sub10}^{ss}(N_{100}) = \frac{Q}{\beta(d_p)N_{100} + WL(d_p) + DL} \tag{2}$$

With a constant $Q$, the evolution of $N_{sub10}$ can be solved analytically. For the decay rate approach, $Q$ is zero and its

corresponding $N_{sub10}$ is

$$N_{sub10}(t) = N_{sub10}^{ini} \cdot \exp\{-[\beta(d_p)N_{100} + WL(d_p) + DL] \cdot t\} \tag{3}$$

where $N_{sub10}^{ini}$ is the concentration at $t = 0$.

**2.4 Coagulation coefficient**

The measured coagulation coefficient was retrieved from the steady-state concentration or the decay rate of sub-10 nm particles using the box model. With a constant $Q$, $N_{sub10}$ as a function of a varying $N_{100}$ is

$$\frac{N_{sub10}^{ss}(N_{100})}{N_{sub10}^{ss}(0)} = 1 - \frac{N_{100}}{N_{100} + \frac{WL(d_p) + DL}{\beta(d_p)}} \tag{4}$$

For the steady-state concentration approach, $\beta(d_p)$ was retrieved by fitting Eq. 4 to the measured $N_{sub10}^{ss}(N_{100})$. For the decay rate approach, $\beta(d_p)$ was retrieved by fitting Eq. 3 to the measured $N_{sub10}$ as a function of $t$. Both approaches require predetermined values of $WL(d_p)+DL$, which can be readily determined using the decay rate approach.

The expression of the theoretical Brownian coagulation coefficient is given in Eq. 5,

$$\beta(d_1, d_2, \alpha, A) = 2\pi(d_1 + d_2)(D_1 + D_2) \cdot \frac{1 + Kn}{1 + 0.377Kn + \frac{4}{3\alpha}Kn(1 + Kn)} \cdot E(A) \tag{5}$$

where $\beta$ is the coagulation coefficient (cm$^3$ s$^{-1}$) between two particles with the sizes (nm) of $d_1$ and $d_2$, $D_1$ and $D_2$ are particle

diffusivity (m$^2$ s$^{-1}$), Kn is the Knudsen number (-) of particles determined by their sizes and mean free path (Fuchs and Sutugin, 1971), $\alpha$ is the mass accommodation coefficient characterizing the effectiveness of coagulation (-), $A$ is the Hamaker constant (J), and $E$ is a multiplicative factor characterizing the influence of van der Waals attractive force on particle coagulation. The value of the Hamaker constant $A$ used to calculate $E$ was taken from previous literature on NaCl particle coagulation (Okuyama et al., 1984) and H$_2$SO$_4$-NH$_3$ condensation (Stolzenburg et al., 2020) or obtained by fitting Eq. 5 to the measured data. The

value of $E(A)$ was obtained using a model accounting for the van der Waals attractive force and the methods to compute $E(A)$ can be found in previous studies (Sceats, 1989; Chan and Mozurkewich, 2001). We took the multiplicative factor from this model instead of the van der Waals coagulation coefficient to avoid the minimal differences among the coagulation coefficients given by different models (Lehtinen and Kulmala, 2003; Sceats, 1989; Ouyang et al., 2012) at $A = 0$ J. With $A = 0$ J and correspondingly $E = 1$, $\beta$ is reduced to the hard-sphere coagulation coefficient (Lehtinen and Kulmala, 2003). In Eq. 5, particles



are assumed to be spherical and their density is assumed to be equal to the bulk density. Therefore, the empirical $E(A)$ may also account for the influence of particle shape on the $\beta$ of e.g. NaCl particles (Okuyama et al., 1984, 1986).

## 3 Results and discussion

### 3.1 Steady-state concentration approach

This approach was used to measure the CoagS of $NH_4HSO_4$ and NaCl particles. Figure 2 shows a typical experimental run using 4 nm particles. At the beginning of the experiment, monodisperse 4 nm particles were injected into the chamber and their number concentration ($N_4$) at the chamber inlet was then maintained constant until the end of the experiment. The $N_4$ in the chamber reached a plateau after ~0.5 h, indicating that the 4-nm particle source was dynamically balanced with wall loss and dilution. After that, 100 nm particles were injected into the chamber and their concentration increased slowly towards a

high steady-state concentration. As shown in Figure 2, $N_4$ in the chamber decreased significantly with an increasing $N_{100}$. Since the source (i.e., particle injection) and other sinks (i.e., wall loss and dilution) of 4 nm particles were kept constant, the decrease of $N_4$ must be caused by the increasing CoagS. At $t$ = ~5.7 h, the inlet $N_{100}$ was set to zero. As a result, $N_{100}$ in the chamber decreased following an exponential decay curve (Eq. 3) and $N_4$ in the chamber increased correspondingly.

===================

190                                     Figure 2

===================

The measured $N_{sub10}$ in the chamber during the increase and decrease of $N_{100}$ was approximated as the steady-state concentration. The validity of this approximation is guaranteed by the short residence time of sub-10 nm particles compared to that of 100 nm particles (e.g., 8.6 min seconds for 4 nm particles versus 65 min for 100 nm particles), which was verified

using the box model. Data when $N_{100}$ in the chamber changed quickly was excluded from the analysis for better accuracy.

Using the box model in Eq. 4, we retrieved size-dependent $(WL(d_p)+DL)/\beta(d_p)$ from the measured steady-state $N_{sub10}$ as a function of $N_{100}$. As shown in Figure 3, with the CoagS contributed by $1.2 \times 10^4$ cm$^{-3}$ 100-nm particles, the pseudo-steady-state $N_{sub10}$ decreased by half compared to the case with zero CoagS. The measured decreasing trend of $N_{sub10}$ with an increasing $N_{100}$ is very consistent with a curve fitted using Eq. 4, with $(WL(d_p)+DL)/\beta(d_p)$ as the fit parameter. The value of $\beta(d_p)$ is then

obtained using a predetermined value of $WL(d_p)+DL$ as depicted below.

===================

Figure 3

===================

### 3.2 Decay rate approach

This approach was used to measure the decay rate (i.e., total loss rate) of particles due to wall loss, dilution, and coagulation sink. As shown in Figure 4, the exponential decay of monodisperse particle concentration in the chamber can be well characterized using the box model (Eq. 3). When measuring $WL(d_p)+DL$, only sub-10 nm particles or 100 nm particles were injected into the chamber, and particle losses due to self-coagulation were negligible. Combining $WL(d_p)+DL$ and the $(WL(d_p)+DL)/\beta(d_p)$ retrieved using the steady-state concentration approach, we obtained the size-dependent $\beta(d_p)$ of $NH_4HSO_4$

and NaCl particles.

===================





Figure 4

===================

Figure 4 also shows that WL($d_p$)+DL was strongly size-dependent. For 100 nm particles whose decay was mainly driven by

dilution, the measured decay rate was close to the theoretical dilution rate calculated using chamber volume and the total flow rate through the chamber. For sub-10 nm particles, wall loss governed the decay rate and it increased quickly as $d_p$ decreased. The size dependency of WL($d_p$) is similar to that of $\beta(d_p)$. For this reason, even though CoagS was expected to increase with a decreasing $d_p$, there was no obvious dependence of the relationship between $N_{sub10}$ and $N_{100}$ on $d_p$ in Figure 3.

The decay rate approach was also used to determine the CoagS of OOM particles. As shown in Figure 5, with $2.6 \times 10^4$ cm$^{-3}$

100 nm particles, $N_{sub10}$ decayed substantially faster than the rate contributed by only wall loss and dilution. According to the box model (Eq. 3), $\beta(d_p)$ was calculated using the difference between the decay rates fitted to the experiments with and without 100 nm particles.

===================

Figure 5

===================

When there were 100 nm particles in the chamber, $N_{sub10}$ could only be measured using the SMPS that focused on the sub-10 nm size range. Although the SMPS data was in good consistency with the CPC data, the temporal resolution of SMPS was limited by its low overall detection efficiency. To reduce statistical uncertainties of the decay rate approach, each experiment was performed at least three times such that the measured difference between the decay rates was sufficient to provide a

relatively accurate estimate of $\beta(d_p)$.

### 3.3 Coagulation coefficient

By comparing the measured coagulation coefficients to theoretical values, we demonstrate that the CoagS of the test 3-10 nm particles was effective, i.e., almost every collision between one 3-10 nm particle and one 100 nm particle contributed to CoagS and the mass accommodation coefficient $\alpha$ for coagulation scavenging was near unity. As shown in Figure 6, the measured

coefficients were comparable to the theoretical coefficients calculated with $\alpha = 1.0$. In contrast, the theoretical curve calculated with $\alpha = 0.1$ was approximately one order of magnitude lower than the measured values. This large discrepancy was far beyond that can be explained by measurement uncertainties. Hence, the measured coagulation sink must be effective.

Further, we found that the van der Waals force contributed to the CoagS of test particles. This contribution is usually accounted for in terms of the collision between vapors, clusters, and small particles, for which the multiplicative factor $E$ in Eq. 5 typically

ranges from 2 to 8 (Halonen et al., 2019; Stolzenburg et al., 2020; Okuyama et al., 1984). The empirical Hamaker constants for the condensation of H$_2$SO$_4$-NH$_3$ (Stolzenburg et al., 2020) vapors and the coagulation between NaCl particles (Okuyama et al., 1984, 1986) were estimated to be $4.6 \times 10^{-20}$ J and $8.9 \times 10^{-20}$ J, respectively. Assuming these values are also valid for calculating the CoagS of sub-10 nm particles, the theoretically predicted $E$ was ~1.4. As shown in Figure 6, the van der Waals model could explain the measured coagulation coefficients better than the hard-sphere model, indicating the contribution of

the van der Waals force to CoagS.

===================

Figure 6

===================

We also estimated the value of empirical Hamaker constants by fitting a coagulation coefficient curve to experimental results

(Figure 6). The best fit Hamaker constants for NH$_4$HSO$_4$, NaCl, and OOM particles were $1.6 \times 10^{-19}$ J, $2.5 \times 10^{-19}$ J, $9.0 \times 10^{-20}$ J,





respectively. The differences between the coagulation coefficients calculated using the best fit Hamaker constant (e.g., $2.5\times10^{-19}$ J for NaCl) and the previously reported value ($8.9\times10^{-20}$ J) (Okuyama et al., 1984, 1986) were minor (< 15 %).

No significant size dependence of the effectiveness of the CoagS was observed for the test sub-10 nm particles. With particle size being decreased from 10 nm to 3 nm, the value of coagulation coefficient increased because of the increasing particle diffusivity (Figure 6). However, the effectiveness of coagulation as indicated by the ratio between the measured coagulation coefficients and theoretical predications was relatively constant. In addition, we did not observe a significant size dependence of the effectiveness of particle wall loss to the Teflon surface, which provides circumstantial evidence for the effectiveness of CoagS contributed by scavenging particles.

Particle composition had only a minor influence on the CoagS of the test particles. As shown in Figure 6, the measured coagulation coefficients of $NH_4HSO_4$, NaCl, and OOM particles were close to each other. More importantly, this closure implies that the effectiveness of CoagS was not affected by the composition of the test particles. The measured minor influence of particle composition is supported by the theoretical prediction that the coagulation coefficient is a weak function of composition-related properties such as particle density. For instance, there is a small (7 %) difference between the theoretical hard-sphere coagulation coefficients for $NH_4HSO_4$ and NaCl particles. Coincidentally, this small difference is further compensated by the different Hamaker constants for van der Waals coagulation coefficients.

### 3.4 Atmospheric implications

The measured effective CoagS of the test particles is strong evidence for the effectiveness of the CoagS of atmospheric new particles. In terms of particle composition, $NH_4HSO_4$ particles were used to represent atmospheric new particles formed by acid-base nucleation (Chen et al., 2012) and growth (McMurry et al., 2005) and OOM particles were used to represent organics nucleation and growth(Bianchi et al., 2016). Recent studies have shown that sulfuric acid (with stabilizing base and water) is a major contributor to new particle formation and OOMs may also contribute to new particle growth in polluted urban environments (Yao et al., 2018; Cai et al., 2021c; Qiao et al., 2021). Hence, the test inorganic and organic particle composition cover that of typical atmospheric new particles, though the latter is expected to be more complex. Further, the test OOM particles were unstable against evaporation in the chamber because the surrounding gaseous OOMs had been removed. Despite this, they were scavenged effectively by 100 nm particles as well as the Teflon chamber wall. Hence, it is unlikely that growing OOM particles surrounded by supersaturated gaseous OOMs in the atmosphere are not effectively scavenged by coagulation. In terms of particle size, the chamber experiments provide evidence for the effective CoagS of 3-10 nm particles. Previous studies (Cai et al., 2021c; Deng et al., 2021) have reported the effectiveness of CoagS of sub-1.5 nm acid-base clusters formed in NPF events. The 1.5-3 nm size range was not covered in this study and the effectiveness of CoagS in this size range remains to be explored.

Concerning the effectiveness of CoagS, there are most likely to be other reasons for the frequent and intensive NPF events in polluted environments. To further investigate the reasons, we calculate the survival probability of new particles, which governs the occurrence of NPF events (McMurry et al., 2005; Cai et al., 2021b). The theoretical survival probability was predicted using Eq. 6 (Lehtinen et al., 2007),

$$P(d_1 \rightarrow d_2) = \exp\left\{-\frac{1}{m-1}\left[1-\left(\frac{d_1}{d_2}\right)^{m-1}\right] \cdot d_1 \cdot \frac{\text{CoagS}(d_1)}{\text{GR}}\right\} \tag{6}$$

where $P$ is the survival probability, $d_1$ is 3 nm, $d_2$ is 10 nm, and $m$ is $-\text{dlnCoagS/dln}d_p$. The value of $m$ is determined as -1.7 according to the measured particle size distributions. Due to the strong size-dependence of CoagS, new particles are mainly scavenged at small sizes and hence we focus on the survival of sub-10 nm particles in this study. When GR is size-dependent, we calculate $P$ by discretizing $[d_1, d_2]$ into many size bins and then multiplying the survival probabilities for each bin. The





CoagS was calculated using measured particle size distributions (Kulmala et al., 2012a). For the convenience of discussion, we use the condensation sink (CS) of sulfuric acid below to quantify the size-dependent CoagS, yet it is worth being clarified that the effectiveness of CS of condensable vapors (Krechmer et al., 2017; Tuovinen et al., 2020) is different from the effectiveness of CoagS.

The value of GR was retrieved from the measured particle size distributions using two methods. One is the mode-fitting method that tracks the temporal evolution of the peak diameter of the new particle mode (Kulmala et al., 2012a). The other is the appearance time method with coagulation correction that tracks the appearance of particles at different sizes (Cai et al., 2021a). Both methods have been used to estimate the GR in urban Beijing (Deng et al., 2020; Qiao et al., 2021), though there is a systematic difference in the GRs retrieved using these two methods. For instance, the GR of sub-5 nm particles estimated using the appearance time method was ~3 times that of the GR estimated using the mode-fitting method for the same dataset. Due to measurement uncertainties, it was previously difficult to infer the accurateness of these two methods by comparing the retrieved GR and the theoretical GR contributed by gaseous precursors (Qiao et al., 2021). Using the CoagS and particle survival probability, we provide more information on the value of GR, as will be discussed in detail below. In addition to the retrieved GR, we also calculate the GR contributed by gaseous sulfuric acid.

The occurrence of NPF events in urban Beijing was significantly suppressed by the high CoagS. As shown in Figure 7, the value of CS during NPF events in urban Beijing usually ranged from 0.003 to 0.03 $s^{-1}$ and a few NPF events were observed at a high CS up to 0.07 $s^{-1}$. Such a high CS is approximately one order of magnitude higher than the CS in relatively clean environments such as Finnish boreal forest (Cai et al., 2017b). As a result, it substantially suppressed particle survival, as indicated by the low survival probability and the low NPF frequency at high CS (e.g., > 0.02 $s^{-1}$) in Figure 7.

==================

Figure 7

==================

Comparing the observed NPF events and the predicted survival probability indicates a moderate or high (e.g., > 3 nm $h^{-1}$) growth rate of new particles in urban Beijing. As shown in Figure 7, most of the observed NPF events, as well as the decreasing NPF frequency as a function of CS, can be explained with a moderate median GR of 3 nm $h^{-1}$. The NPF events observed with CS > 0.03 $s^{-1}$ may be associated with high GRs and/or particle formation rates. In contrast, with a low GR of 1 nm $h^{-1}$, the theoretical survival probability is as low as 1 % at CS = 0.009 $s^{-1}$. Considering the typical new particle formation rate in urban Beijing (~10 $cm^{-3}$ $s^{-1}$ for 3 nm particles) (Kulmala et al., 2022), such a low survival probability cannot explain a majority of the observed NPF events.

The moderate GR provides insights into new particle growth mechanisms. Sulfuric acid (with its stabilizing bases) is known to contribute to new particle growth for its low volatility (Stolzenburg et al., 2005; Stolzenburg et al., 2020). To provide a sufficient particle survival probability (equivalent to a 3 nm $h^{-1}$ GR), sulfuric acid concentration needs to be as high as ~1.5×$10^7$ $cm^{-3}$ (Figure 7). For urban Beijing, such a high concentration can only be occasionally observed in summer (Deng et al., 2020), in which season sulfuric acid concentration is the highest due to the strongest solar radiation (Qiao et al., 2021). Hence, it can be implied that although sulfuric acid is an important precursor for new particle growth, there should be other gaseous precursors (such as OOMs) contributing to the growth and survival of new particles.

Extending the above discussions from 3-10 nm to sub-3 nm provide hints for the growth mechanisms of sub-3 nm particles. The median GR of sub-3 nm particles in urban Beijing was previously reported to be ~1 nm $h^{-1}$, which was consistent with the growth rate contributed by gaseous sulfuric acid with a median concentration of ~5×$10^6$ $cm^{-3}$ (daily maximum value) (Deng et al., 2020). As a result of this consistency, sulfuric acid and its stabilizing bases were previously thought to govern the growth of sub-3 nm new particles in urban Beijing (Deng et al., 2020; Qiao et al., 2021). However, assuming an effective CoagS of





sub-3 nm particles, it can be concluded that a moderate GR (e.g., 3 nm h$^{-1}$) of sub-3 nm particles is also needed to explain the observed NPF events (see Figure S1 in the Support Information), i.e., other undetected gaseous precursors or unrevealed mechanisms may also contribute majorly to sub-3 nm particle growth.

The moderate GR of sub-3 nm particles inferred from the survival probability is supported by the appearance time method, which reported a 3 nm h$^{-1}$ median GR for the same dataset (Qiao et al., 2021). We also find that the appearance time method

tends to report a more accurate GR of sub-3 nm particles than the mode-fitting method, because the mode-fitting method may be influenced by the constantly forming new particles (see Figures S2 and S3 for illustration).

In addition to particle survival probabilities, the effectiveness of CoagS also solidifies the understanding of atmospheric new particles based on aerosol kinetics. For example, new particle formation rate is usually calculated using population balance equations (Kulmala et al., 2012b; Cai and Jiang, 2017), and the net coagulation sink term is the governing term for NPF in

polluted environments (Deng et al., 2021; Cai and Jiang, 2017). If CoagS was overestimated by one order of magnitude, particle formation rate would be correspondingly overestimated by the same magnitude. By measuring the effectiveness of CoagS, we demonstrate the accuracy of the calculated particle formation rates. Another example is that with a relatively accurate survival probability guaranteed by the effectiveness of CoagS, one can evaluate the influences of NPF on air quality by estimating the surface area and mass concentrations of grown new particles.

**4 Conclusions**

To investigate the survival of atmospheric new particles especially in polluted environments with high coagulation sinks, we experimentally determine the effectiveness of the coagulation sink of 3-10 nm particles with chamber experiments. Monodisperse particles composed of ammonium bisulfate, sodium chloride, and oxygenated organic molecules were generated to represent typical new particles formed in the atmosphere. Their pseudo-steady-state concentration and decay rate were

measured as a function of the concentration of 100 nm particles, which was used to scavenge the 3-10 nm particles. The coagulation coefficient between 3-10 nm particles and 100 nm particles was subsequently determined using a box model. We found that the measured coagulation coefficient increased sharply with a decreasing particle size, whereas it was not sensitive to the chemical composition of the test 3-10 nm particles. By comparing the measured coefficients with theoretical values, we demonstrated that the coagulation sink of the test 3-10 nm particles was effective with a near-unity mass accommodation

coefficient. Further, the measured coefficient suggests that van der Waals force contributes to the coagulation sink and results in a ~40 % higher value than hard-sphere coagulation. These experiments indicate that the coagulation sink of atmospheric 3-10 nm new particles was unlikely to be substantially overestimated. Hence, the low theoretical survival probabilities of new particles contrasting to the new particle formation events observed at high coagulation sinks should be caused by underestimated growth rates. We show that a median growth rate of new particles at 3 nm h$^{-1}$ can explain most of the measured

new particle formation events in urban Beijing. This 3 nm h$^{-1}$ growth rate is usually higher than the growth rate contributed by gaseous sulfuric acid. Hence, the measured effective coagulation sink indicates that in addition to gaseous sulfuric acid, other gaseous precursors also contribute to the growth of new particles.





**Code and data availability**

The chamber data for particle coagulation and the codes for the box model are available upon request from the corresponding
author.

**Author Contribution**

R.C. and J. Kangasluoma designed the research; R.C. and E.H. conducted the chamber experiments; J. Jiang, C.Y. and R.C.
collected the atmospheric data; R.C. and M.K. analyzed the data; R.C. wrote the paper with input from all co-authors.

**Competing interests**

One coauthor, M.K., is a member of the editorial board of Atmospheric Chemistry and Physics.

**Acknowledgements**

This study is funded by the Academy of Finland (project no. 332547, 1325656, 346370), the National Science Foundation of
China (22188102), Samsung PM$_{2.5}$ SRP, the Vilho, Yrjö, and Kalle Väisälä Foundation, "Quantifying carbon sink,
CarbonSink+ and their interaction with air quality" INAR project funded by Jane and Aatos Erkko Foundation, and European
Research Council (ERC) project ATM-GTP (contract No. 742206). We thank Mikael Ehn, Erkki Siivola, and Qiang Zhang
for helpful discussions on the design of the experimental setup. We also thank Hannu H Koskenvaara and Pasi Aalto for
technical supports.

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



## Figures

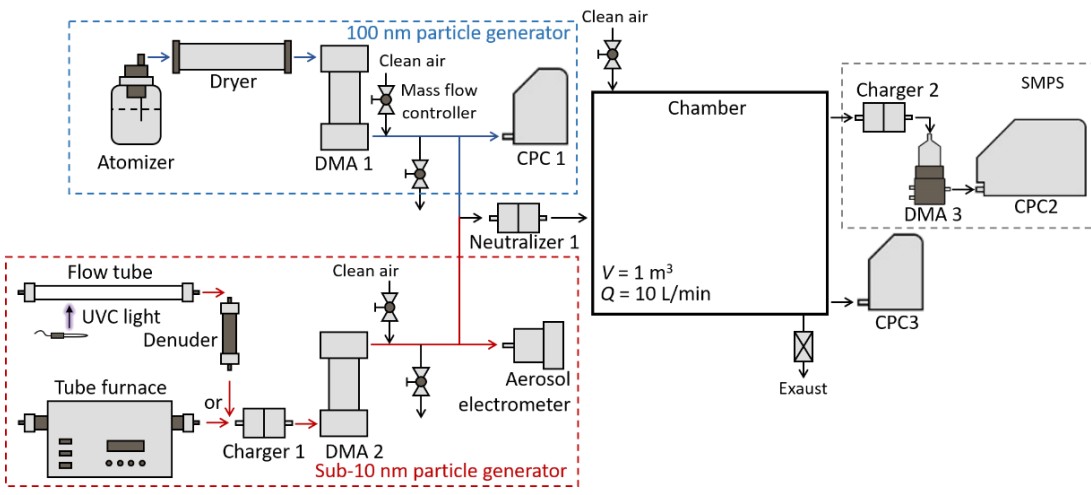

**Figure 1**. Schematic of the chamber experiment setup for measuring size-dependent coagulation coefficient

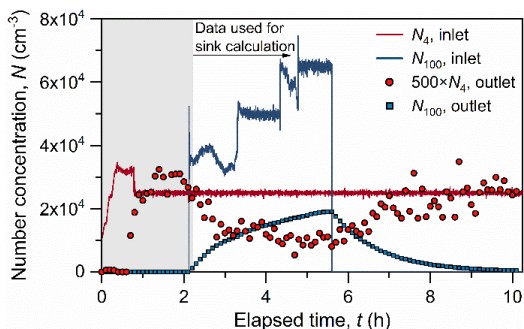

**Figure 2**. Time series of particle number concentrations measured at the inlets and outlets of the chamber during an experimental run. The inlet particle concentrations ($N$) were controlled using mass flow controllers. Assuming perfect mixing

in the chamber, particle concentrations in the chamber are equal to the outlet concentrations. The subscript of $N$ indicates particle size in nm. Due to the long residence time (~1 h) of 100 nm particles, the outlet $N_{100}$ changed slowly towards its steady-state concentration when the inlet $N_{100}$ was kept stable.

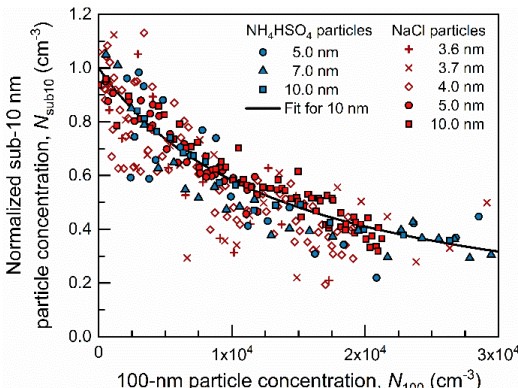

**Figure 3**. The normalized pseudo-steady-state concentration of monodisperse sub-10 nm particles ($N_{sub10}$) as a function of 100 nm particle concentration ($N_{100}$). $N_{sub10}$ is normalized by dividing it by the value measured at $N_{100} = 0$. The curve is obtained by fitting Eq. 4 to the measured data.

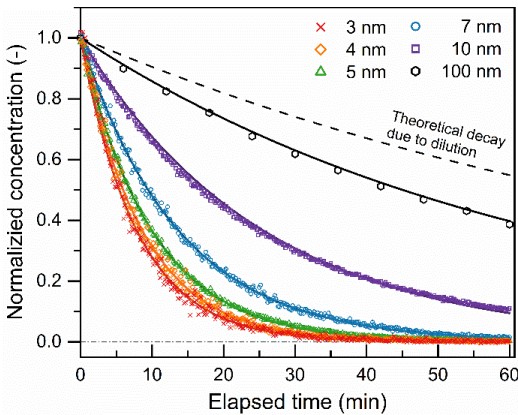

**Figure 4**. Decay of the concentration of $NH_4HSO_4$ and NaCl particles in the chamber due to wall loss and dilution. Particle concentration is normalized by dividing it by the value measured at $t = 0$. Curves are obtained by fitting Eq. 4 to the measured data.



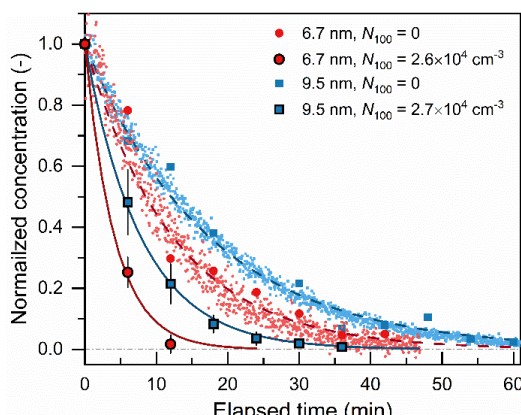

**Figure 5**. Decay of the concentration of sub-10 nm organic particles in the chamber due to wall loss, dilution, and coagulation sink. Sub-10 nm particle concentration is normalized by dividing it by the value measured at $t = 0$. $N_{100}$ is the concentration of 100 nm particles used to provide the coagulation sink. The small and large markers represent data measured by a CPC and an SMPS, respectively. The variation bar indicates the standard deviation of normalized sub-10 concentration in repeated experiments.

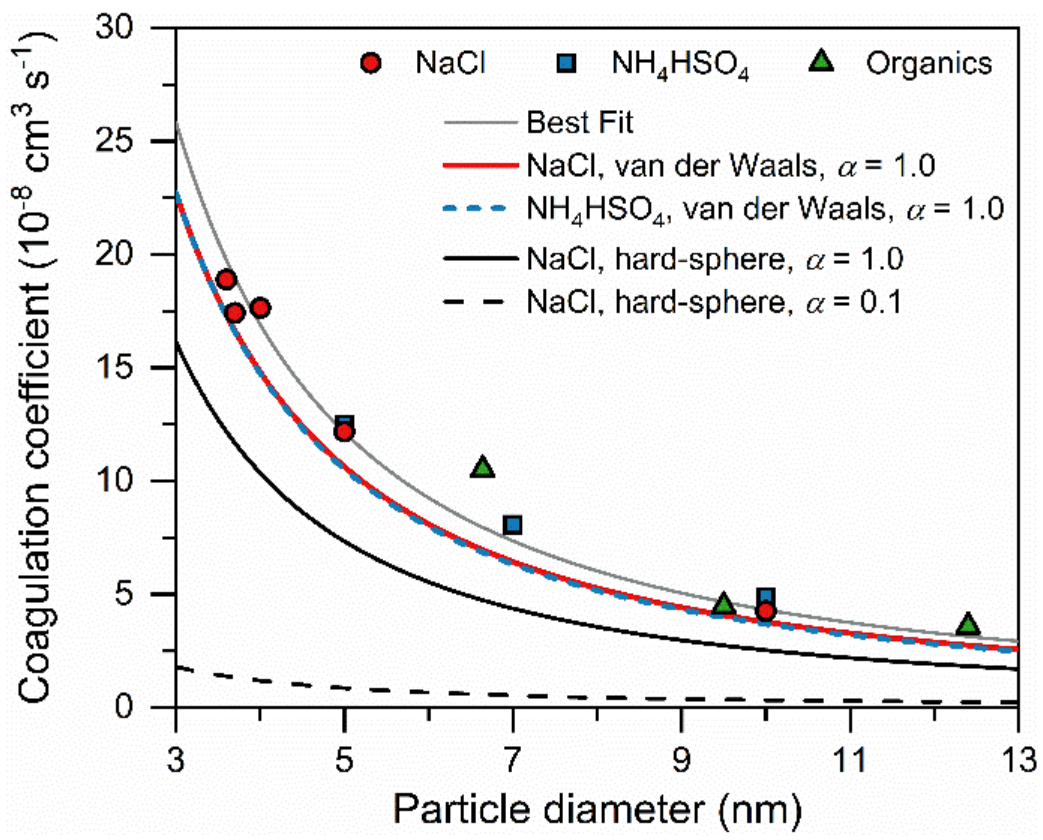

**Figure 6**. Measured and theoretical coagulation coefficients between sub-10 nm particles and 100 nm particles. The hard-sphere coagulation coefficient only accounts for particle diffusion. The van der Waals coagulation coefficient also accounts for the van der Waals attractive force. $\alpha$ is the mass accommodation coefficient characterizing the effectiveness of coagulation. The best fit is obtained by fitting the best empirical Hamaker constant to the measured size-dependent coagulation coefficient.

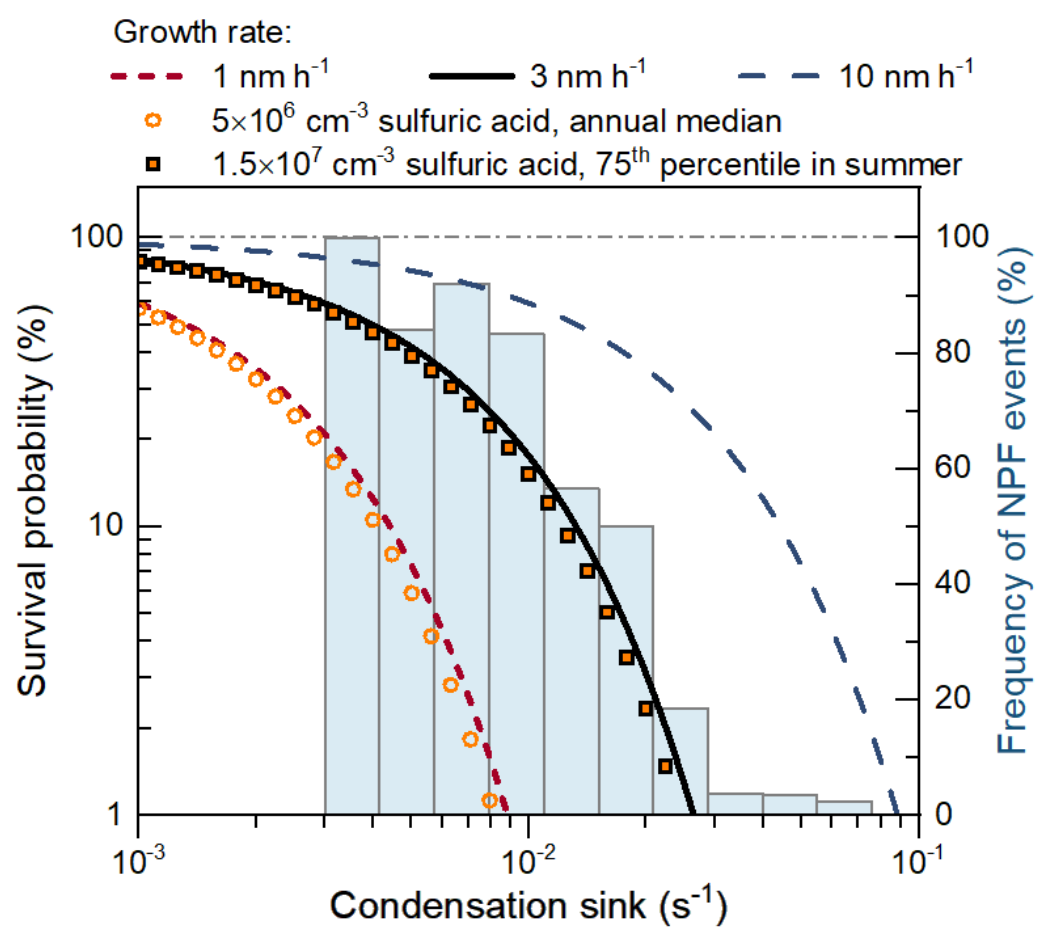

**Figure 7**. The theoretical survival probability of new particles and the measured frequency of new particle formation events in urban Beijing as a function of the condensation sink. The survival probability is calculated for 3-10 nm particles assuming an effective coagulation sink, i.e., the mass accommodation coefficient is equal to 1.0. The condensation sink characterizes the size-dependent coagulation sink during each new particle formation event. The NPF frequency is determined as the ratio of NPF days to all the measurement days within the given condensation sink range. This figure shows that a 1 nm $h^{-1}$ growth rate (or $5\times10^{6}$ $cm^{-3}$ sulfuric acid) is insufficient to explain the measured new particle formation in urban Beijing.