# Peer review of "The effectiveness of coagulation sink of 3-10 nm atmospheric particles"

_Atmospheric Chemistry and Physics, 2022_

## Referee Comment (RC2)

Cai et al focuses on evaluation of the coagulation rate coefficient between sub-10 nm particles and 100 nm particles in a chamber environment. In reading the manuscript, I have concerns on the manner the experiments were conducted (a large fraction of 100 nm particles remain charged after neutralization, and this is not considered in results) and and analysed (equation 5 is not a valid approach in the transition regime). I have the following comments for the authors to address:

1. Figure 1. It appears that after DMA selection, the authors sent particles through a bi-polar charger (labelled neutraliser, but the authors should note the type of neutraliser in terms of source type, and strength). For the 100 nm particles, roughly 47% of them are charged at steady-state (near 20% will be +1 net charge, near 20% will be -1 net charge, and 7% multiply charged). Therefore, it seems the experiment is not between neutral particles exclusively, but between a population of charged 100 nm particles and sub 10 nm particles, as well as uncharged 100 nm particles and sub 10 nm particles. It is not clear if the charged particle-uncharged particle potential is negligible in this instance, and in comparison to the often weaker effect of van der Waals potentials, it is likely not negligible. Without directly addressing the charge influence (or clarifying that charged particles were removed), it is not clear to me that the measurements and analysis reported are sufficiently done to merit publication, as they would not necessarily be representative of the atmosphere.

2. Section 2.1. The mixing within the chamber needs to be described in much greater detail. One concern in the experiments is that the inlet sub-10 nm particle concentration is orders of magnitude higher than the outlet concentration. The inlet flow is not diluted infinitely quickly upon entering the chamber, and there is certainly a region near the inlet where the sub 10 nm concentration is similar to the 100 nm particle concentration (their injected concentrations are similar)

3. Methods (General). The authors need to more explicitly state the number concentration ranges of the 100 nm particles used and the sub-10 nm particles.

4. Equation (1). The authors neglect the influence of sub-10 nm particle self-coagulation on the differential equation governing the sub-10 nm particle concentration. From the limited data in the main text, it appears that sub-10 nm particle concentrations in the chamber were much lower than the 100 nm particle concentrations, but this needs to be discussed more directly, and brief calculations to show it is negligible.

5. Alongside neglecting the potential influence of 100 nm particle charge on the coagulation rate, equation (5) in the manuscript is not an appropriate way to handle coagulation in the transition regime accounting for potential interactions. In the continuum limit (very small Kn), it is acceptable to write that the coagulation coefficient using the formula:
$$\beta_{Cont} = 2\pi(d_1 + d_2)(D_1 + D_2)E_{Cont} \qquad \text{(R1)}$$
Similarly, in the free molecular limit (very large Kn), one can write:
$$\beta_{FM} = \left(\frac{\pi kT}{2m_{12}}\right)^{1/2}(d_1 + d_2)^2 E_{FM} \qquad \text{(R2)}$$
where $k$ is Boltzmann's constant and $m_{12}$ is the reduced mass of the two particles. However, $E_{Cont}$, which can be calculated for spheres exactly using the approach of Fuchs for continuum collisions, and $E_{FM}$, which is calculated using an approach by Fuchs & Sutugin, Ouyang, or Sceats, are not one and the same, and for the van der Waals potential, usually $E_{FM} > E_{Cont}$. Because in the transition regime, the transport features controlling collision are evolving as particle diameter increases (Kn decreases), it is not correct to express equation (5) as a single enhancement factor applicable to all particles (the enhancement evolves as well). The authors should instead adopt an approach to fit $E_{Cont}$ AND $E_{FM}$ to results. The Ouyang (2012) reference of the manuscript attempts to do this, as does Fuch's limiting sphere approach when potentials are included and Sceats 1989 (I believe the Sceats approach is the one used in

Stolzenburg 2020; see equation 3 in their paper). When the authors state they used the methods from previous studies it is not clear if they mean $E_{Cont}$ or $E_{FM}$, and only considering one is not correct in the transition regime. As inference of this parameter is the central purpose of this manuscript, this issue combined with neglecting charge effects (again, unless charged particles were removed) is concerning.

6. Equation (5), continued. The authors need to explicitly state how they are defining Kn (define lambda, the mean free path in term of other, better defined parameters, as this is not the gas mean free path). They should then discuss the Kn range examined in this study, to state whether their results apply to coagulation in the continuum limit, the free molecular limit, or a transition regime range. There is a chance Kn (which tends to be lower for particle-particle collisions) is sufficiently low that they are close to the continuum limit, such that they can simply use a continuum limit coagulation expression and $E_{Cont}$. However, as per comment 5 if the coagulation range studied is in the transition regime, then I am afraid the authors need to rethink their analysis.

---

## Author Comment (AC1)

**Responses to Reviewers' Comments on Manuscript ID ACP-2022-262**

(The effectiveness of coagulation sink of 3–10 nm atmospheric particles)

We thank the reviewers for their efforts and comments that help to improve this manuscript. The reviewers' comments are addressed in the following paragraphs and the manuscript has been revised accordingly. More details on the experiments and analysis have been provided to facilitate understanding.

The comments are shown as sans-serif blue texts and our responses are shown as serif black texts. Changes are highlighted in the revised manuscript and shown as "quoted underlined texts" in the responses. References are given at the end of the responses.

**Reviewer #1**

The paper by Cai et al. aims to solve one of the most confusing problems related to particle survival by quantifying coagulation coefficient. Direct experimental evidence for the effectiveness of the CoagS of new particles was provided by well controlled chamber experiments. Comparison between the measured coefficient with theoretical predictions shows that almost every coagulation leads to the scavenging of one particle, and the coagulation sink exceeds the hard-sphere kinetic limit due to van der Waals attractive force. Base on the measurement, the authors proposed high theoretical survival probabilities of new particles in NPF events observed at high coagulation sinks are caused by high growth rates. I suggest the publication of this paper. One issue should be addressed in the abstract "....the measured coagulation coefficient increases significantly with a decreasing particle size...". Please specify for what particle size this conclusion refers to. Obviously, your statement is completely wrong because coagulation coefficient with 10 nm particle decreases with the decreasing size of coagulating particles.

**Response**: We thank the reviewer for these comments. This sentence was revised as "… the measured coagulation sink of 3-10 nm particles increases significantly with a decreasing particle size…".

The reviewer is correct that the coagulation coefficient is a function of both particle sizes. We use the coagulation sink instead for simplicity and emphasize these particles are sub-10 nm particles to avoid confusion. Although the scavenger particles (100 nm in this study) are not given in the revised sentence, this statement should be valid for the atmosphere as the coagulation sink is majorly contributed by particles around 100 nm.

**Reviewer #2**

*Cai et al focuses on evaluation of the coagulation rate coefficient between sub-10 nm particles and 100 nm particles in a chamber environment. In reading the manuscript, I have concerns on the manner the experiments were conducted (a large fraction of 100 nm particles remain charged after neutralization, and this is not considered in results) and analysed (equation 5 is not a valid approach in the transition regime). I have the following comments for the authors to address:*

**Response**: We appreciate the reviewer's deep insights into the experiments and analysis. The concerns are addressed below following the comments. Briefly, we find that the concerns are related to the presentation rather than the experiments or analysis. Accordingly, we clarified that 100 nm particles remained charged after neutralization and they resemble atmospheric particles in terms of the steady-state charge fraction. The charge fraction of particles are emphasized when discussing the coagulation coefficient. For Eq. 5, we clarified that the enhancement factor therein is a function of particle size and it is valid for all regimes.

*1. Figure 1. It appears that after DMA selection, the authors sent particles through a bi-polar charger (labelled neutraliser, but the authors should note the type of neutraliser in terms of source type, and strength). For the 100 nm particles, roughly 47% of them are charged at steady-state (near 20% will be +1 net charge, near 20% will be -1 net charge, and 7% multiply charged). Therefore, it seems the experiment is not between neutral particles exclusively, but between a population of charged 100 nm particles and sub 10 nm particles, as well as uncharged 100 nm particles and sub 10 nm particles. It is not clear if the charged particle-uncharged particle potential is negligible in this instance, and in comparison to the often weaker effect of van der Waals potentials, it is likely not negligible. Without directly addressing the charge influence (or clarifying that charged particles were removed), it is not clear to me that the measurements and analysis reported are sufficiently done to merit publication, as they would not necessarily be representative of the atmosphere.*

**Response**: Thanks. We clarified that the 100 nm particles were charged at steady-state so that their charge fraction resembles the charge fraction of ambient particles. We also clarified that the interaction between a charged particle and a neutral particle may influence the coagulation coefficient, and hence the measured coagulation coefficient is an empirical value characterizing the coagulation sink of new particles scavenged by background particles at steady-state charge fraction.

We added a few sentences to the Methods section to clarify the experiments and the fact that 100 nm particles were at steady state charge distribution, representing atmospheric conditions. After the generation and DMA production, we used a bi-polar charger to neutralize both sub-10 nm particles and 100 nm particles. The neutralized particles were sent to the chamber via a Teflon tube. For sub-10 nm particles, most of these particles could be neutralized and the Teflon tube can effectively remove the remaining charged particles via electrostatic losses. For 100 nm particles, as the reviewer has pointed out, a considerable fraction of them was charged and the charged particles could pass through the Teflon tube. As a result, 100 nm particles in the chamber were charged at a steady state instead of being neutral.

We clarified that the particles with steady-state charge fractions are representative of atmospheric particles. A recent study has demonstrated the feasibility to measure the particle size distributions accurately using an SMPS without a charger because atmospheric particles are charged at a steady state (Li et al., 2022). We used neutral sub-10 nm particles in the

experiments and clarified the reason that "most (>90 %) sub-10 nm particles are neutral at a steady state charge distribution". This also minimize the influence of the Coulomb force on the CoagS. In contrast, we think that the 100 nm particles after the bi-polar charger can better characterize the coagulation sink compared to neutral particles.

We agree with the reviewer that the interaction between a charged particle and a neutral particle may influence the coagulation coefficient. In the revised Theory and Results sections, we added "Besides, the interaction between a charged particle and a neutral particle may influence the coagulation coefficient, which is also included in the experimentally determined $E(A)$ in this study." and "This enhancement is likely due to the van der Waals force and potentially the interaction between a neutral sub-10 nm particle and a charged 100 nm particle."

We also tried to compute the coagulation enhancement due to the interaction between a charged particle and a neutral particle. Using the methods in Santos et al. (2019), we found this enhancement is possibly weak: For the coagulation between one neutral 3-10 nm particle and one doubly charged 100 nm particle, the multiplicative enhancement factor is 1.00003-1.0007. Although this method was proposed for the kinetic regime only and there might be uncertainties in the results, it indicates that interaction between a charged particle and a neutral particle may be minor. Nevertheless, the intertaction between charged and neutral particles would not affect our conclusions that the coagulation sink of new particles is effective and slightly above the hard-sphere. More importantly, the charge fraction of 100 nm particles in the experiments were representative of atmospheric particles.

The type of neutralizer has been specified as "A neutralizer (Ni-63, 90 MBq)…" in the revised manuscript.

2. Section 2.1. The mixing within the chamber needs to be described in much greater detail. One concern in the experiments is that the inlet sub-10 nm particle concentration is orders of magnitude higher than the outlet concentration. The inlet flow is not diluted infinitely quickly upon entering the chamber, and there is certainly a region near the inlet where the sub 10 nm concentration is similar to the 100 nm particle concentration (their injected concentrations are similar)

**Response**: We added a sentence to clarify that "A low-speed fan placed near the bottom center of the chamber was used to mix particles in the chamber and its mixing ability had been experimentally validated". The experiments were to test the response of outlet particle concentration to the inlet particle concentration (similar to the experiment in Fig. 2). We found that the outlet particle concentration followed the theoretical curve, and the decay rate of 100 nm particles due to dilution and wall loss was close to the theoretical decay rate due to dilution. This indicates that the mixing time for particles is much smaller than the residence time; hence the mixing is rapid enough for this experiment.

It is true that the sub-10 nm particle concentration should be high in the inlet region of the chamber. However, we controlled the inlet particle concentration ($<10^5$ cm$^{-3}$) such that the self-coagulation of sub-10 nm particles was negligible. For instance, with a collision enhancement, the coagulation sink of sub-10 nm particles contributed by self-coagulation is no larger than $0.5 \times 3.3 \times 10^{-9}$ cm$^3$ s$^{-1} \times 10^5$ cm$^{-3} = 1.7 \times 10^{-4}$ s$^{-1}$, corresponding to a residence time of ~100 min. Compared to the loss rate or the residence time of sub-10 nm particles (see Fig. 4), it is reasonable to conclude that the experiments were negligibly affected by self-coagulation.

3. Methods (General). The authors need to more explicitly state the number concentration ranges of the 100 nm particles used and the sub-10 nm particles.

**Response**: We stated the concentration range was $10^4$-$10^5$ cm$^{-3}$. Typical concentrations of particles has also been presented in Fig. 2.

4. Equation (1). The authors neglect the influence of sub-10 nm particle self-coagulation on the differential equation governing the sub-10 nm particle concentration. From the limited data in the main text, it appears that sub-10 nm particle concentrations in the chamber were much lower than the 100 nm particle concentrations, but this needs to be discussed more directly, and brief calculations to show it is negligible.

**Response**: Thanks. In the revised Methods section, we added "The concentrations of 100 nm and sub-10 nm particles were controlled to be $10^4$-$10^5$ cm$^{-3}$ such that self-coagulation was negligible ($< 0.5 \times 3.3 \times 10^{-9}$ cm$^3$ s$^{-1}$ $\times 10^5$ cm$^{-3}$ = $1.7 \times 10^{-4}$ s$^{-1}$) and the CoagS provided by the 100 nm particles were distinguishable against dilution and wall losses." As the self-coagulation of $10^5$ cm$^{-3}$ inlet particles was negligible, the self-coagulation rate of particles in the chamber would be even lower.

5. Alongside neglecting the potential influence of 100 nm particle charge on the coagulation rate, equation (5) in the manuscript is not an appropriate way to handle coagulation in the transition regime accounting for potential interactions. In the continuum limit (very small Kn), it is acceptable to write that the coagulation coefficient using the formula:

$\beta_{Cont}$=$2\pi$ $(d_1+d_2)(D_1+D_2)E_{Cont}$ (R$_1$)

Similarly, in the free molecular limit (very large Kn), one can write:

$\beta_{FM}$=$(\pi kT/2m_{12})^{1/2}(d_1+d_2)_2 E_{FM}$ (R$_2$)

where $k$ is Boltzmann's constant and m$_{12}$ is the reduced mass of the two particles. However, $E_{Cont}$, which can be calculated for spheres exactly using the approach of Fuchs for continuum collisions, and $E_{FM}$, which is calculated using an approach by Fuchs & Sutugin, Ouyang, or Sceats, are not one and the same, and for the van der Waals potential, usually $E_{FM}$>$E_{Cont}$. Because in the transition regime, the transport features controlling collision are evolving as particle diameter increases (Kn decreases), it is not correct to express equation (5) as a single enhancement factor applicable to all particles (the enhancement evolves as well). The authors should instead adopt an approach to fit $E_{Cont}$ AND $E_{FM}$ to results. The Ouyang (2012) reference of the manuscript attempts to do this, as does Fuch's limiting sphere approach when potentials are included and Sceats 1989 (I believe the Sceats approach is the one used in Stolzenburg 2020; see equation 3 in their paper). When the authors state they used the methods from previous studies it is not clear if they mean $E_{Cont}$ or $E_{FM}$, and only considering one is not correct in the transition regime. As inference of this parameter is the central purpose of this manuscript, this issue combined with neglecting charge effects (again, unless charged particles were removed) is concerning.

**Response**: We would like to argue that Eq. 5 should be valid for the transition regime because the Kn term is exactly used to correct for the transition regime (and the free molecular regime). In the revised manuscript, we clarified that "The expression of the theoretical Brownian coagulation coefficient covering the free molecular, transition, and continuum regimes is given in Eq. 5…" Alternative equations for all the three regimes can be found elsewhere, e.g., in Ouyang et al.

(2012) and Table 13.1 in Seinfeld and Pandis (2006) and. As shown in the figure below, the curve for Eq. 5 with $E = 1$ is comparable with other formulae and it approaches the $\beta_{Cont}$ and $\beta_{FM}$ as Kn approaches 0 and infinite, respectively.

[Figure]

The reviewer is correct that $E$ is a function of particle sizes rather than a constant. $E(A)$ in Eq.5 has been correspondingly revised as $\underline{E(A,d_1,d_2)}$ to emphasize that it is "underline{size-dependent}". In this study, we compute the size-dependent $E(A)$ using the approaches in Ouyang et al. (2012) and Alam (1987). When fitting the experimental results, we fitted the empirical Hamaker constant to the measured coagulation coefficients to guarantee the size-dependency of $E(A)$. However, the $E(A)$ was only weakly dependent on the sub-10 nm particle size, e.g., 1.33-1.39 for 3-10 nm ammonium bisulfate particles. This variation is usually smaller than typical uncertainties in atmospheric measurements. Consequently, we report a constant value in the Results and discussion section and clarified that "$E$ was ~1.4 for particles in this size range with a weak dependence on the particle size".

We would also like to clarify that the main purpose is to investigate particle survival by measuring the effectiveness of CoagS of new particles rather than the enhancement. As discussed above, the 100 nm particles charged at the steady-state can better represent atmospheric particles. The main aim is to figure out whether the CoagS deviates from its theoretical value by orders of magnitudes. The results have convincingly shown that the CoagS was effective.

6. Equation (5), continued. The authors need to explicitly state how they are defining Kn (define lambda, the mean free path in term of other, better defined parameters, as this is not the gas mean free path). They should then discuss the Kn range examined in this study, to state whether their results apply to coagulation in the continuum limit, the free molecular limit, or a transition regime range. There is a chance Kn (which tends to be lower for particle-particle collisions) is sufficiently low that they are close to the continuum limit, such that they can simply use a continuum limit coagulation expression and $E_{Cont}$. However, as per comment 5 if the coagulation range studied is in the transition regime, then I am afraid the authors need to rethink their analysis.

**Response**: Thanks, we added Eq. 6-7 to the revised manuscript for Kn and the mean free path. As discussed above, Eq. 5 and the enhancement factor were for all regimes. Accordingly, we emphazied "…the theoretical Brownian coagulation

coefficient covering the free molecular, transition, and continuum regimes…" and "The Kn term is used to correct for the transition and free molecular regimes…".

**Reference**

Alam, M. K.: The Effect of van der Waals and Viscous Forces on Aerosol Coagulation, Aerosol Science and Technology, 6, 41-52, 10.1080/02786828708959118, 1987.

Li, Y., Chen, X., and Jiang, J.: Measuring size distributions of atmospheric aerosols using natural air ions, Aerosol Science and Technology, 56, 655-664, 10.1080/02786826.2022.2060795, 2022.

Ouyang, H., Gopalakrishnan, R., and Hogan, C. J., Jr.: Nanoparticle collisions in the gas phase in the presence of singular contact potentials, J Chem Phys, 137, 064316, 10.1063/1.4742064, 2012.

Santos, B., Cacot, L., Boucher, C., and Vidal, F.: Electrostatic enhancement factor for the coagulation of silicon nanoparticles in low-temperature plasmas, Plasma Sources Science and Technology, 28, 045002, 10.1088/1361-6595/ab0a2b, 2019.

Seinfeld, J. H., and Pandis, S. N.: Atmospheric Chemistry and Physics, 2nd ed., John Wiley & Sons, Inc., New Jersey, 2006.

---

## Referee Report (RR1)

I believe the authors still need to consider and revise the manuscript to address prior comments. The revised submission contains minimal modification to the manuscript and as it stands the authors do not fully clarify the calculation methods they apply, and they do not fully analyze the results from their experiments.

1. Equation 5 is still misleading without more explanation. The enhancement factor, if expressed in this way, would not only be a function of just the Hamaker constant and particle diameters, but also the gas pressure, gas temperature, and composition $E(A, d_1, d_2, kT, P)$. It is actually combination of the free molecular enhancement factor and continuum enhancement factor, and depends on the Knudsen number (really it is $E(A/kT, Kn)$). I recommend in describing calculations, the authors start with theoretically more appropriate expressions (those of Sceats, as shown in Stolzenburg et al, 2020, the equations in Alam 1987, or in Ouyang et al 2012) and then show where $E(A/kT, Kn)$) that they apply comes from. With regards to enhancement factor calculation, the authors only state: "The value of E(A) was obtained using a model accounting for the van der Waals attractive force and the methods to compute size-dependent E(A) can be found in previous studies (Alam, 1987; Sceats, 1989; Chan and Mozurkewich, 2001)." What was the exact equation applied? These references take slightly different approaches from one another. Assuming they are using equation (24) from Alam, 1987 for $E(A/kT, Kn)$), then the authors could clearly show their $E(A/kT, Kn)$ function, as Alam 1987 uses separate enhancement factors for the continuum and free molecular regimes. If they were instead using Chan & Mozurkewich equation (27), which does not show a transition regime enhancement factor explicitly but instead a transition regime coagulation rate, then they would still derive a $E(A/kT, Kn)$ expression with separate continuum and free molecular enhancement factors within it. The authors also need to clarify their method of calculations for the free molecular enhancement factor, was Marlow's equation used, as in Alam 1987 or Sceats's equation as in Chan and Mozurkewich?

   A final note on this topic- the authors state "We took the multiplicative factor from this model instead of the van der Waals coagulation coefficient to avoid the minimal differences among the coagulation coefficients given by different models (Lehtinen and Kulmala, 2003; Alam, 1987; Ouyang et al., 2012) at A = 0 J." This is tangential to the issue with equation (5). The issue with current presentation of equation (5) is that it hides that there are different enhancement factors for different regimes, and for van der Waals potentials, the enhancement factor in the free molecular regime is typically considerably larger than in the continuum regime. Particular care has to be taken to explain what the Knudsen numbers are for the processes examined, and how enhancement factors are calculated.

2. The authors have dismissed comments (3) and (6) from prior review without completely modifying the manuscript. As written, the manuscript (and response to reviewer comments) come off as quite hasty. I recommend that the authors include, perhaps as a supplemental table, the number concentrations of 100 nm particles the initial number concentrations of 3-10 nm particles (ideally broken up into sub ranges, perhaps 1 nm increments) for each experiment performed. The Kn range or average Kn for each bin should also be shown, as shown a best fit $E(A/kT, Kn)$ for that size range. There appear to be a number of potential valuable data in Figure 3, but they are not provided in a way that can be easily used by others.

---

## Author Response (AR2)

**Responses to Reviewers' Comments on Manuscript ID ACP-2022-262**

(The effectiveness of coagulation sink of 3–10 nm atmospheric particles)

We thank the reviewer again for the efforts and comments that help to improve this manuscript. The reviewers' comments are addressed in the following paragraphs and the manuscript has been revised accordingly. We rewrote the method section on the theoretical coagulation coefficient and provided the details in the supporting information. We also added a table to the supplementary information to provide more details on the experiments.

The comments are shown as sans-serif blue texts and our responses are shown as serif black texts. Changes are highlighted in the revised manuscript and shown as "quoted underlined texts" in the responses. References are given at the end of the responses.

I believe the authors still need to consider and revise the manuscript to address prior comments. The revised submission contains minimal modification to the manuscript and as it stands the authors do not fully clarify the calculation methods they apply, and they do not fully analyze the results from their experiments.

Equation 5 is still misleading without more explanation. The enhancement factor, if expressed in this way, would not only be a function of just the Hamaker constant and particle diameters, but also the gas pressure, gas temperature, and composition $E(A,d1,d2,kT,P)$. It is actually combination of the free molecular enhancement factor and continuum enhancement factor, and depends on the Knudsen number (really it is $(A/kT,)$). I recommend in describing calculations, the authors start with theoretically more appropriate expressions (those of Sceats, as shown in Stolzenburg et al, 2020, the equations in Alam 1987, or in Ouyang et al 2012) and then show where $E(A/kT,Kn)$) that they apply comes from. With regards to enhancement factor calculation, the authors only state: "The value of E(A) was obtained using a model accounting for the van der Waals attractive force and the methods to compute size-dependent E(A) can be found in previous studies (Alam, 1987; Sceats, 1989; Chan and Mozurkewich, 2001)." What was the exact equation applied? These references take slightly different approaches from one another. Assuming they are using equation (24) from Alam, 1987 for $E(A/kT,Kn)$), then the authors could clearly show their $E(A/kT,Kn)$ function, as Alam 1987 uses separate enhancement factors for the continuum and free molecular regimes. If they were instead using Chan & Mozurkewich equation (27), which does not show a transition regime enhancement factor explicitly but instead a transition regime coagulation rate, then they would still derive a $E(A/kT,Kn)$ expression with separate continuum and free molecular enhancement factors within it. The authors also need to clarify their method of calculations for the free molecular enhancement factor, was Marlow's equation used, as in Alam 1987 or Sceats's equation as in Chan and Mozurkewich?

A final note on this topic- the authors state "We took the multiplicative factor from this model instead of the van der Waals coagulation coefficient to avoid the minimal differences among the coagulation coefficients given by different

models (Lehtinen and Kulmala, 2003; Alam, 1987; Ouyang et al., 2012) at A = 0 J." This is tangential to the issue with equation (5). The issue with current presentation of equation (5) is that it hides that there are different enhancement factors for different regimes, and for van der Waals potentials, the enhancement factor in the free molecular regime is typically considerably larger than in the continuum regime. Particular care has to be taken to explain what the Knudsen numbers are for the processes examined, and how enhancement factors are calculated.

**Response**: We rewrote Section 4.2 on the theoretical coagulation coefficient to clarify how the theoretical enhancement factor was computed in this study and added computation details in the supplementary information. $E(A, d_1, d_2)$ has been revised as $E(A/kT, \text{Kn})$.

In the revised manuscript, we give the coagulation coefficients for the continuum regime and the free molecular regime,

$$\beta_C = 2\pi(d_1 + d_1)(D_1 + D_2) \cdot E\left(\frac{A}{kT}, 0\right) \tag{8}$$

$$\beta_{FM} = \frac{\pi}{4}\alpha(d_1 + d_1)^2\sqrt{c_1^2 + c_2^2} \cdot E\left(\frac{A}{kT}, \infty\right) \tag{9}$$

We clarified that Eq. 5 can be reduced to Eqs. 8 and 9 as Kn approaches the continuum limit and the free molecular limit, respectively.

We first compute $E(A/kT, 0)$ and $E(A/kT, \infty)$ using the formulae reported in Chan and Mozurkewich (2001), which were fitted to numerical solutions to the formulae in Sceats (1989). We then use the interpolation formula in Alam (1987) to obtain $E(A/kT, \text{Kn})$, as Kn for particles in the chamber experiments were in the transition regime (1.0-1.9). We also compared the interpolation formula in Alam (1987) to those in Sceats (1989) and Ouyang et al. (2012), finding that the difference ($< 4\%$) among these interpolation formulae was smaller than the experimental uncertainties.

The above procedure has been clarified in the manuscript as:

"For the coagulation between 3-10 nm and 100 nm particles, Kn ranges from 1.0-1.9. Consequently, coagulation occurred in the transition regime. To obtain the theoretical $E(A/kT, \text{Kn})$, we first computed $E(A/kT, 0)$ and $E(A/kT, \infty)$ using the formulae reported in Chan and Mozurkewich (2001), which were fitted to the numerical solution to the integral from Sceats (1989). The results were then extended to the transition regime using the interpolation formula in Alam (1987). We also compared $E(A/kT, \text{Kn})$ to the results interpolated using the methods in Sceats (1989) and Ouyang et al. (2012), finding good consistencies among those methods. More details on the theoretical $\beta$ and $E(A/kT, \text{Kn})$ can be found in the Supporting Information (SI)."

In the revised SI, we added Eqs. S1-S17 to clarify the exact equations for $E(A/kT, 0)$, $E(A/kT, \infty)$, and the interpolation formulae for $E(A/kT, \text{Kn})$. A new figure was also added to show the consistency among these formulae and indicate the range of Kn for test particles in the chamber experiments.

[Figure]

**Figure S1**. The van der Waals enhancement factor for the collision between a 1-1000 nm particle and a 100 nm particle. The shaded area indicates the Knudsen number of particles used in the chamber experiments.

2. The authors have dismissed comments (3) and (6) from prior review without completely modifying the manuscript. As written, the manuscript (and response to reviewer comments) come off as quite hasty. I recommend that the authors include, perhaps as a supplemental table, the number concentrations of 100 nm particles the initial number concentrations of 3-10 nm particles (ideally broken up into sub ranges, perhaps 1 nm increments) for each experiment performed. The Kn range or average Kn for each bin should also be shown, as shown a best fit ($A/kT$,) for that size range. There appear to be a number of potential valuable data in Figure 3, but they are not provided in a way that can be easily used by others.

**Response**: We followed the reviewer's suggestions and added a table to the SI, which summarizes the Kn and best fit coagulation coefficient $\beta_{meas}$ for each particle size. The measured ($A/kT$,), which is determined as the ratio of $\beta_{meas}$ to the theoretical hard-sphere collision coefficient, is also given, though their values are not used in the main text.

This table also includes the concentration of sub-10 nm particles in the chamber during each experiment in Figs. 3 and 5. The concentration of 100 nm particles can be found in the horizontal axis or the figure legend. The inlet concentrations are not given because they are not used to determine $\beta_{meas}$ and the inlet concentration of 100 nm particles varied with time, yet we have clarified in the method section that "The concentrations of 100 nm and sub-10 nm particles were controlled to be $10^4$-$10^5$ cm$^{-3}$". The data in Fig. 3 and other figures have been uploaded to an open-access server so that they could be used by other colleagues.

Unfortunately, we might not fully understand the comment "……the initial number concentrations of 3-10 nm particles (ideally broken up into sub ranges, perhaps 1 nm increments) for each experiment performed". The 3-10 nm particles were classified by a DMA and "the aerosol and sheath flow rates of the DMA were 3.5 and 10 L min$^{-1}$, respectively". For each experiment, only particles with sizes closes to the DMA centroid diameter were selected. As can be seen from Table

S1, the total number concentration of the sub-10 nm particles was relatively low. To minimize random uncertainties, we used the total concetration to compute the coagulation coefficient instead of the size-segrated concetration.

In the revised manuscript, we added:

"They were charged and then monodisperse particles with a certain size were classified using a Vienna-type differential mobility analyzer (DMA)."

"For each experiment, we evaluated the CoagS of particles with a certain size. The size-dependency of CoagS was obtained by repeating the experiment for different particle sizes."